# You Don't Need Domain-Specific Data Augmentations When Scaling Self-Supervised Learning

**Théo Moutakanni**
FAIR at Meta
MICS, Université Paris-Saclay
theomoutakanni@meta.com

**Maxime Oquab**
FAIR at Meta

**Marc Szafraniec**
FAIR at Meta

**Maria Vakalopoulou**
MICS, CentraleSupélec,
Université Paris-Saclay

**Piotr Bojanowski**
FAIR at Meta

## Abstract

Self-Supervised learning (SSL) with Joint-Embedding Architectures (JEA) has led to outstanding performances. All instantiations of this paradigm were trained using strong and well-established hand-crafted data augmentations, leading to the general belief that they are required for the proper training and performance of such models. On the other hand, generative reconstruction-based models such as BEIT and MAE or Joint-Embedding Predictive Architectures such as I-JEPA have shown strong performance without using data augmentations except masking. In this work, we challenge the importance of invariance and data-augmentation in JEAs at scale. By running a case-study on a recent SSL foundation model – DINOv2 – we show that strong image representations can be obtained with JEAs and only cropping without resizing provided the training data is large enough, reaching state-of-the-art results and using the least amount of augmentation in the literature. Through this study, we also discuss the impact of compute constraints on the outcomes of experimental deep learning research, showing that they can lead to very different conclusions.

## 1   Introduction

Self-supervised learning (SSL) has significantly improved performance across various tasks. Despite not requiring supervision, SSL heavily depends on carefully selected data augmentations [9, 11–13, 35]. Previous experimental literature has shown that removing even one augmentation such as random rescaling or color jittering reduces linear evaluation performance on ImageNet1k [15, 41] by at least 10 points [1, 11, 39]. But reliance on image-specific data augmentations limits the generalizability of self-supervised approaches. How can this be applied to graph, time-series or, for example, medical imaging with totally different channels and characteristics? According to multiple studies [1, 5, 21, 37], the best data augmentations are dependent on the target tasks, and no set of hand-crafted data augmentations can solve them all. But is this the case?

The problem is the following: SSL methods that do not build on such augmentations are based on image reconstruction, and are able to achieve competitive results, though only through fine-tuning ([4, 25]). On the other hand, joint-embedding architecture (JEA) methods, delivering strong results without fine-tuning and producing linearly separable features, seem largely reliant on data augmentations in light of the seminal SimCLR [11, 12] study. I-JEPA [2] was the first method to challenge this assumption, but the authors still show a significant performance gap when compared to their augmentation-based alternatives. Therefore, one can reasonably believe that the core principles driving JEA methods towards great performance are inherently dependent on data augmentation,

38th Conference on Neural Information Processing Systems (NeurIPS 2024).

Table 1: **Comparison of our model** trained using `RandomCrop`, without random resizing nor other photometric augmentations against SSL models that do not leverage hand-crafted augmentations. All other models are reconstruction based, in the pixel space or in the latent space, and *use more augmentations* than our setup. We only use `RandomCrop` without resizing, and masking.

| Method | Data | Arch. | ImageNet-1k | | | Other Classif. | | Segm. | Depth |
| | | | val | V2 | real | Places205 | iNat18 | ADE20k | NYUd ↓ |
| --- | --- | --- | --- | --- | --- | --- | --- | --- | --- |
| AIM | DFN-2B+ | H/14 | 70.0 | 43.7 | 64.4 | 51.7 | 64.0 | 13.2 | 0.862 |
| BEIT | INet-1k | L/16 | 73.5 | 59.0 | 78.3 | 57.9 | 40.8 | 6.4 | 0.544 |
| MAE | INet-1k | H/14 | 78.0 | 66.5 | 83.7 | 56.0 | 42.2 | 33.3 | 0.482 |
| I-JEPA | INet-1k | H/14 | 79.3 | 66.0 | 83.8 | 58.4 | 47.6 | 31.1 | 0.537 |
| I-JEPA | INet-22k | g/16 | 71.3 | 59.7 | 78.6 | 59.1 | 55.3 | 32.0 | 0.521 |
| Ours | INet-22k | L/14 | **84.0** | **74.8** | **88.6** | **65.1** | **75.1** | **43.5** | **0.411** |

and the underlying invariance learning that it entails. As a side-effect, JEA setups have been extensively tuned to work with augmentations [9, 13, 35], cementing this belief in the community. In this context, the present paper successfully disproves the assumption that these augmentations are necessary. We show that it is possible to train **state-of-the-art** joint embedding architectures without augmentations other than crops **without random rescaling**, and optionally masking. To the best of our knowledge, we train such model using the **least amount of data-augmentations in the literature**. Data cropping without rescaling, along with masking, might be considered two of the only universal data augmentations that can be applied on a broad number of modalities by just using the properties of the data. We show that building such SoTA model simply requires a larger amount of diverse data, a longer training schedule, and more careful optimization.

For most methods, data augmentations prevent model collapse on trivial solutions such as color histogram [1, 7]. They can also enhance performance by artificially increasing the diversity and size of datasets using multiple crops per image [9, 10]. While preliminary work has attempted to explain the theory behind data augmentations and self-supervised learning or have conducted experimental analysis of their impact and necessity in SSL [1, 5, 11, 21, 37], such analysis has **not been conducted on large-scale SSL pretraining** with newer methods, restricting the conclusions that can be drawn from their results.

In this work, we aim to determine the role of data augmentations at scale by training models using DINOv2 [35] on multiple datasets of varying sizes (from 1M to 140M images), on multiple model sizes (from Small to Large), and using diverse data augmentation combinations.

**Contributions.** By doing a rigorous experimental analysis through this paper, we are the first to prove two counter-intuitive claims that go against most previous assumptions:

**(i)** We show that the impact of data-augmentations invariance enforcement in self-supervised learning is secondary to the impact of their artificial increase in dataset size and distribution.

**(ii)** We explore when and why we can remove hand-crafted data augmentations with minor impact on performance by looking at all scaling aspects of deep learning: data, compute and model size.

In the process, we also provide two by-products on top of our claims:

**(iii)** We show through our experimental setup that scaling laws are ethereal and that practitioners optimizing for different compute and data budget might find very different conclusions.

**(iv)** We achieve state-of-the-art performance on extensive evaluations for a model trained without hand-crafted augmentations. This result is an **existence proof** that joint-embedding architecture methods do not inherently rely on learning invariances defined by data augmentations, challenging the prior beliefs in the community and prompting a new theoretical understanding of SSL.

## 2 Related Work

**Data augmentation in SSL practice.** Data augmentations have been a key component of most self-supervised works, including the early Exemplar-CNN [18]. The network would be trained

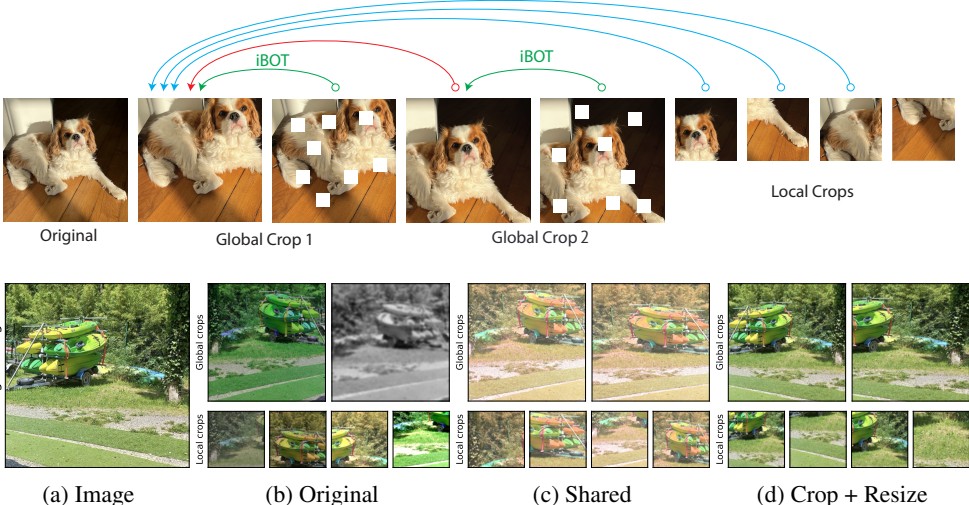

Figure 1: **Top: Visual description of pretraining losses.** In blue: the local to global DINO loss, in red: the global to global DINO loss and in green the latent masked token prediction (iBOT) loss. **Bottom: Our different augmentation strategies.** 'Original' uses several augmentations (`RandomResizedCrop`, `ColorJitter`, `RandomGrayscale`, `GaussianBlur`, `RandomHorizontalFlip` and `RandomSolarize`), 'Shared' uses the same augmentations but shares them between each view of the same image obtained with `RandomResizedCrop`. The 'Crop + Resize' setting only uses `RandomResizedCrop`. We also introduce a 'Crop' setup which uses `RandomCrop` without random rescaling and that is visually similar to 'Crop + Resize'.

using a classification objective, with each class corresponding to data augmentations of one image in the dataset. In the context of early pretext-task-based SSL, when training convolutional neural networks by predicting relative location of patches, Doersch, Gupta, and Efros [16] have observed trivial solutions. The network was overfitting to chromatic aberrations, and the authors used data augmentations to counter this effect. In a similar way, Bojanowski and Joulin [7] have observed that their discriminative model would emphasize trivial factors of variation such as colour. The solution proposed in this work was to discard color information and take the output of Sobel filtering as an input to the algorithm, along with image jitter and flipping. In 2018, the community observed a jump in performance when evaluating on Pascal VOC 2007 image classification [8, 22], the implementation of both works leveraging `RandomResizedCrop` instead of `RandomCrop`. The authors of SimCLR [11] carried out the first extensive ablation of the impact of data augmentation on performance, and many of the conclusions from this work *serve as standard in follow-up SSL work* [10, 24, 26, 35]: most modern joint-embedding self-supervised models learn using at least one loss term that pushes together the representation of two augmented views of the same image. By applying different data augmentations to the two views, they force the model to map the augmented characteristics to the same point, enforcing invariance in the model representation.

**Sensitivity of learning to data augmentation.** With the widely accepted view of the importance of invariance learning for SSL models, several works have studied the impact of the choice of data augmentations on downstream performance: Xiao et al. [52] show that the wrong choice of augmentations can have a detrimental effect; Tian et al. [46] consider data augmentations as a way to reduce the mutual information between different views of an image to improve results. Bendidi et al. [5] show through experiments that the choice, amplitude and combination of transformations effects the learnt representation performance in downstream tasks and that the benefit of data-augmentations depend on each specific class. Purushwalkam and Gupta [37] show that a poor choice of cropping parameters can harm performances.

At the same time, self-supervised learning algorithms based on image reconstruction don't use many data augmentations. Models trained to reconstruct missing patches directly in the pixel space [4, 25, 33] have led to significantly lower linear-evaluation performance but did not leverage any photometric or hand-crafted augmentations. El-Nouby et al. [32] show that BEIT is more robust to smaller dataset size. Recently, I-JEPA [2], based on reconstruction in the feature space obtained strong performance

without using photometric augmentations. The authors split the comparison against other SSL frameworks according to the type of augmentations used (see Table 1 in [2]), suggesting data augmentations make for an unfair advantage in SSL training. It is worth noting that *reconstruction based methods still use random resizing* in their augmentations.

**Theoretical studies.** Furthermore, a series of theoretical works have been studying the *apparently critical* data augmentation for SSL algorithms. For example, Kügelgen et al. [30] prove, under some assumptions, that data augmentations allow isolating content from style, when using InfoNCE as an objective, as in SimCLR. Eastwood et al. [19] follow up on this work, and propose using a structured data augmentation framework in order to disentangle content from style in the learnt representations; the goal is to avoid discarding style information during the learning process. Zhang and Ma [53] propose splitting the multiple operations involved in data augmentation pipelines in order to produce a hierarchy of augmentations: invariance to each augmentation is translated into a loss applied at different stages of the backbone, leading to accuracy improvements. Notably, they mention that *"The most indispensable process in contrastive learning is the data augmentation module."* Saunshi et al. [42] propose seeing data augmentations as inductive biases applied to the learning algorithm, and derive theorems valid in the case of linear representations, in order to further understand the principles driving the success of contrastive learning.

In this work, we provide a critical result regarding the importance of data augmentation, that dramatically improves our understanding of the core principles of self-supervised learning, prompting renewed theory: **data augmentations are not necessary and only cropping without resizing is sufficient to train strong SSL joint-embedding architecture models**.

# 3 Approach

The goal of this work is to study the importance of data augmentation for training joint embedding architectures. In this section, we provide a quick recap of the SSL training algorithm that we use for our study. We then provide a precise description of the experimental setup, with details about the training regimes and data augmentation strategies.

## 3.1 Short description of the DINOv2 algorithm

We tailor our study around the algorithm that was used for training the recently proposed DINOv2 [35] models. We hypothesize that learnings from this study should apply to other SSL methods, but we chose to focus our experimental work on the best modern JEA method. That way, we can provide a strong counter-argument to the assumptions used in previous literature. We briefly summarize the components of the DINOv2 learning algorithm, and for more details, we point to the original publication [35]. Figure 1 provides a schematic diagram of the loss terms used for training the model.

**Global DINO loss.** The DINOv2 training loop is largely based on the DINO algorithm [9]. Given a student ($s$) and a teacher ($t$) network, and two views of the same image $v_1$ and $v_2$, the student network is trained by minimizing a loss between the two representations:

$$\ell(s(v_1), t(v_2)). \tag{1}$$

The teacher network, is taken as a moving average of the student. We refer the reader to [9] for more details about the definition of $\ell$.

**Local reconstruction-based iBOT loss.** DINOv2 models were trained using an additional local masked-image reconstruction loss. As opposed to MAE or BEIT, the reconstruction task in iBOT happens in the latent space, not in the pixel input space. Given a view $v_1$ as used in the DINO loss, we create an impaired version $\tilde{v}_1$ by replacing a fixed percentage of patches with a `[MASK]` token. The target is to reconstruct the value of masked features using the remaining context:

$$\ell(s(\tilde{v}_1), t(v_1)). \tag{2}$$

If several views are used, the iBOT loss can be applied to every such view. The strategy for choosing the masked patches is a hyper-parameter of the method.

**Multicrop and data augmentations.** In practice, one does not use only two views $v_1$ and $v_2$. Following [10], DINOv2 uses Multicrop, with two global crops and eight local crops. Global crops are obtained using `RandomResizedCrop` with a `scale` parameter of $[0.32, 1.0]$, then resized to 224 pixels. For local crops we used $[0.05, 0.32]$ resized to 98 pixels instead. Each crop, be it global

or local, undergoes a series of photometric augmentations: `ColorJitter`, `RandomGrayscale`, `GaussianBlur`, `RandomHorizontalFlip` and `RandomSolarize`. We refer to the official DINOv2 repository for the exact implementation details.[1]

## 3.2 Experimental setup

**Hyperparameters and training regimes.** The original DINOv2 repository proposes two sets of hyperparameters for training SSL models. The first set (that we refer to as the *low-compute regime*) corresponds to a setup for fast experimental iterations, designed to run for 100 epochs (125k iterations) with a batch size of 2048. This setup is optimized for performance on the ImageNet-1k dataset (corresponding to the *low-data regime*). The second set (*high-compute regime*) is designed for longer training runs of 500 epochs (625k iterations) and is optimized for performance on larger datasets, such as ImageNet-22k (the *high-data regime*). These two sets differ in many values, including notably the learning rate schedule and warmup, weight decay, batch size, patch size. In our experiments, we use both sets to compare the behavior of the algorithm against data augmentation. We provide a detailed empirical discussion of the differences in Sec. 4.3.

**Modifications applied on data augmentation.** In our study, we apply different configurations of data augmentations. We provide here the details of the four configurations that we consider.

- **Original**: we use all the data augmentations, exactly like in the original DINOv2 work. Each crop is passed through a series of random photometric augmentations independently.

- **Shared**: each view of an image is obtained using a different crop, but apply the exact same photometric augmentation to all views. Two images in the batch will have different photometric augmentations applied to them.

- **Crop + Resize**: we create views with no photometric augmentations at all, only `RandomResizedCrop`.

- **Crop**: we create views without any photometric augmentations at all. We also don't use `RandomResizedCrop`. Instead, we use a `Resize` to $256 \times 256$ pixels and `Crop` to 224 or 98 pixels, leading to views with the exact same aspect ratio and zoom, with no random resizing. This setup is very close to a masking strategy where different sets of tokens would be dropped, up to pixel alignment to patch boundaries.

Figure 1 provides an illustration of the images that we obtain by applying these configurations. We omit the 'Crop' setting from the illustration given that the 'Crop' and 'Crop+Resize' provide images that are hard to qualitatively differentiate with a naked eye, even if they differ in term of final performances. In the end, our 'Crop' strategy only applies two augmentations between generated views: cropping without random resizing and random patch masking, which are two very similar augmentations (the cropping drops pixels outside of a field-of-view, and masking inside of it). We chose to keep those two specific augmentations as we think that they can be generalized to nearly all kind of data, from graphs to 3D medical images and time-series.

**Implementation details.** For our study, we use the standard ImageNet-1k [41] and ImageNet-22k [15] datasets, as well as the LVD-142M dataset originally used in DINOv2 [35]. The selection of these datasets allows us to benchmark the impact of their size in the trainings. The pre-training code performs 100 epochs in 27 hours on 5 compute nodes of 8 A100-80GB GPUs each, where 1 epoch is set to 1250 iterations. For evaluations, we follow DINOv2 linear evaluation protocol on multiple different tasks including classification (ImageNet-1k [41], Places205[55], ImageNet-v2 [38], ImageNet-Real [6], iNaturalist'18 [49], FlickrLogos32 [40], GTSRB [45], RP2K [36], Products-10k [3], FireRisk [43], RESISC [14], MIMIC [28], CheXpert [27], VinDr [31], NIH-14 [51]), segmentation (ADE20k [55]) and depth estimation (NYU-Depth v2 [44]).

Table 2: **New domains classification results** of DINOv2 ViT-L trained on ImageNet-22k when varying data augmentations. None of those domains were used in the pretraining data of the models.

| Domain (Metric) | Remote Sensing (Acc.) | | Medical Imaging (AUROC) | | | |
| Dataset | FireRisk | RESISC | MIMIC | CheXpert | VinDr | NIH-14 |
|---|---|---|---|---|---|---|
| Original | 59.0 = | 91.9 = | 75.3 = | 83.3 = | 80.0 = | 72.5 = |
| Shared | 60.8 ↑ 1.8 | 91.6 ↓ 0.2 | **75.8** ↑ 0.5 | **85.6** ↑ 2.3 | 81.0 ↑ 1.0 | **74.4** ↑ 1.9 |
| Crop+Resize | 60.7 ↑ 1.7 | 91.1 ↓ 0.8 | **75.8** ↑ 0.5 | 84.1 ↑ 0.8 | 80.8 ↑ 0.8 | 73.7 ↑ 1.2 |
| Crop | **60.9** ↑ 1.9 | **92.3** ↑ 0.4 | 75.5 ↑ 0.2 | 83.0 ↓ 0.3 | **82.5** ↑ 2.5 | 74.2 ↑ 1.7 |

# 4 Experiments and discussion

## 4.1 Why do we need augmentations?

The first question we want to answer is: why do we need data augmentations? Are they systematically needed, because they constitute a core component of the modeling? Are they a trick that facilitates the optimisation of deep neural networks, exactly like in supervised learning?

**The role of data-augmentations.** In the context of supervised learning, data augmentations have been used as a means of virtually augmenting the training dataset [21]. That way, additional training samples could be obtained, reducing the risks of overfitting and improving the robustness of the models trained. In self-supervised learning however, data-augmentations have been usually modeled as a way to enforce representations to be invariant toward specific image's characteristics, such as color or scale [46, 52, 53].

**Invariance can be harmful.** Most of the self-supervised methods and their data-augmentations have been optimised on ImageNet. While DINOv2 has probably been optimised on more benchmarks, some tasks and domain are missing. In Table 2 we show the results of removing hand-crafted data-augmentations on new domains (not used for training) and in Table 3 (left) on new tasks (in the training set but not used for hyperparameter search). We can see that data-augmentations are actually harmful on most of those, meaning that understanding the real impact of augmentations and invariance while removing them is very important to increase SSL robustness.

**Disambiguating the impact of data augmentation in self-supervised learning.** We developed the 'Shared' configuration that still applies augmentations to artificially increase the number of training samples but *doesn't enforce invariance* to those augmentations. We compare invariance in Table 3 (right) and show that the 'Shared' setup has effectively lower invariance than the 'Original' one. We quantify it using the average cosine similarity of 16 augmented views of the same image, repeated on 100 images from different classes. Higher cosine similarity means higher invariance. We also report a mean/std normalized similarity, computing the metrics using negative pairs' cosine similarity.

**Enforcing invariance losses its usefulness at scale.** By comparing the 'Shared' and 'Original' results in Fig. 2, we can see that both setups reach way above $80\%$ linear-probe accuracy on ImageNet1k when trained on all three datasets. While the 'Original' setup gets better performances, we can see

---
[1] https://github.com/facebookresearch/dinov2 (Apache-2.0 license)

---

Table 3: **(left): New task classification results** of DINOv2 ViT-L trained on ImageNet-22k when varying data augmentations. None of those tasks were used to tune DINOv2's hyperparameters. **(right): Measure of invariance toward augmentation**. Higher cosine similarity means higher invariance as the model embeds multiple augmentations of the same image to closer vectors.

| Task | Logos | Signs | Products | | | Metric | Cosine | Normalized |
| Dataset | FlickrLogos32 | GTSRB | RP2K | Products-10k | | | Similarity | Cos. Sim. |
|---|---|---|---|---|---|---|---|---|
| Original | 80.1 = | 80.3 = | 92.8 = | 66.5 = | | Original | **0.69** | **2.69** |
| Shared | 78.2 ↓ 1.9 | 78.9 ↓ 1.4 | **93.4** ↑ 0.6 | **68.1** ↑ 1.6 | | Shared | 0.64 | 2.62 |
| C+R | **80.8** ↑ 0.7 | 76.3 ↓ 4.0 | 93.3 ↑ 0.5 | 67.7 ↑ 1.2 | | C+R | 0.58 | 2.47 |
| Crop | 76.9 ↓ 3.2 | **85.1** ↑ 4.8 | **93.4** ↑ 0.6 | 67.3 ↑ 0.8 | | Crop | 0.56 | 2.44 |

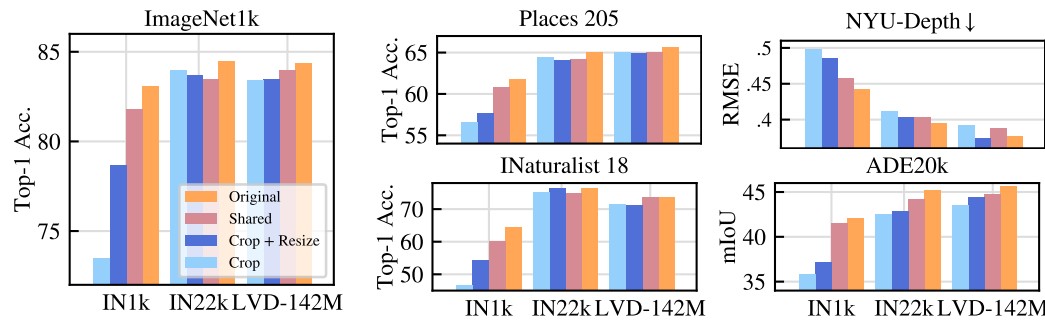

Figure 2: **Impact of dataset size when varying data augmentations.** Results of ViT-L on linear evaluation benchmarks, including classification (ImageNet1k, Places 205 and INaturalist18), depth estimation (NYU-Depth) and segmentation (ADE20k). Cropping without resizing ('Crop') reaches very high performances on a wide variety of benchmarks, given that the dataset size is large enough.

that the gap decreases with the amount of data, from $-1.3\%$ when trained on ImageNet1k and $-1.0\%$ with ImageNet22k, to $-0.4\%$ with LVD-142M. As we show in Fig. 3 (left), the gap between the 'Shared' and 'Original' setups also decreases with the number of training epochs. According to experiments in prior literature [1, 11, 39], data augmentations are required at the sample level to enforce invariance and obtain competitive performance. For example, when trained and evaluated on ImageNet1k, SimCLR and BYOL respectively lose 27.6 and 13.1 points of accuracy in the 'Crop+Resize' setup [39]. In contrast, our 'Shared' setup only loses $1.2\%$, disproving this view.

**Increasing the sample count is the key to strong SSL.** We know that reconstruction-based models like BEIT, MAE or I-JEPA don't need handcrafted data-augmentations on ImageNet1k, and according to El-Nouby et al. [32], *"denoising autoencoders are more sample efficient than joint embedding techniques"*. Those findings led us to conjecture that the real impact of data-augmentation is to artificially increase the number of samples and allow JEA to reach good performance with less data.

To assess this, we compare the 'Crop + Resize' and the 'Shared' settings. Both settings do not enforce invariance, but they differ because the 'Shared' method artificially increases the sample count of the pretraining dataset. We show in Fig. 2 that the gap with respect to the 'Original' setting is way bigger for 'Crop + Resize' setting the compared to the 'Shared' one when we use ImageNet1k to pretrain our models. Without the sample count increase of data-augmentations, DINOv2 reaches lower performance on all five benchmarks when pretrained on a small dataset (Fig. 2).

**Data-augmentations play the same role in supervised and self-supervised learning.** All those results lead to the same conclusion: invariance is useful but not necessary for the learning of JEA, and augmentations artificially increase the number of samples in the pretraining dataset. This also explains why reconstruction-based methods do not need as many augmentations: they are more data efficient [32]. But this is not their only role, as data-augmented setups perform better in shorter trainings (Fig. 3, left). Since setups eventually converge to the same result with sufficient data, this

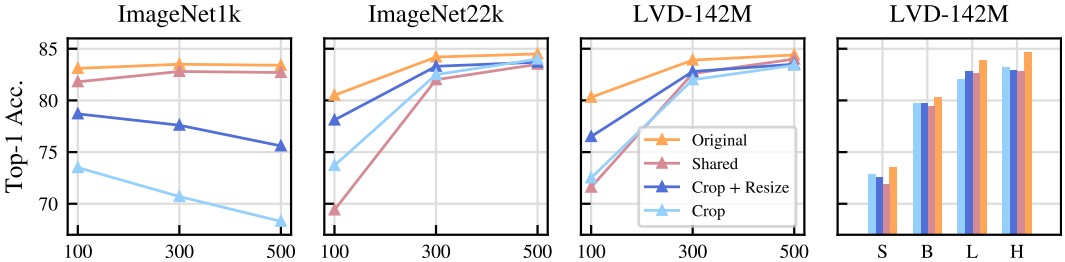

Figure 3: **Impact of data augmentations when we scale the number of training epoch (left) or the ViT architecture size (right)** on the accuracy of a linear probe on ImageNet1k for a ViT-L when pretraining on ImageNet1k, ImageNet22k and LVD-142M. The 'Original' and 'Shared' setups scale with the number of epochs for all datasets, but the 'Crop' and 'Crop+Resize' setups only scale with larger datasets.

suggests data augmentations are likely linked to easier optimization rather than to a core learning signal. They are just another hyper-parameter.

## 4.2 Can we remove hand-crafted augmentations totally?

We focused on what was the impact of data augmentations in modern self-supervised learning using the 'Original', 'Shared' and 'Crop + Resize' settings. We will now show that we can reach state-of-the-art performance without hand-crafted data-augmentations and resizing.

**Scaling dataset size.** Given that DINOv2 is more robust than other methods when training on ImageNet1k without handcrafted data-augmentations [39], and knowing from Sec. 4.1 that data augmentations mainly increase artificially the dataset size, we hypothesize that a self-supervised JEA can reach SOTA performance by increasing the size of the pretraining dataset used, even when removing the random resizing. We can see in Fig. 2 that increasing the dataset size has a notable property: all four settings converge to the same performances and the gap almost disappears between settings with and without hand-crafted augmentations, displaying a non-decreasing scaling curve with the number of samples. The linear-probe accuracy gap decreases below $1\%$ on ImageNet1k and remains small on other benchmarks.

**Scaling pretraining epochs.** The highest performances were obtained when we pretrained our models for 500 epochs on larger datasets and 100 on the smaller one. However, a question arises: what happens if we train for longer on ImageNet1k, and faster on ImageNet22k and LVD-142M. In Fig. 3 (left) we show that the behavior of low and high data regimes differ. While the larger datasets show non-decreasing scaling curves with respect to the number of epochs, ImageNet1k has an uncoupling between the settings with and without data-augmentations. We notice that training longer on smaller scale dataset is harmful for performances when we don't use the dataset size increase property of data-augmentations. This is probably a consequence of overfitting and another proof of our first claim in Sec. 4.1. What is more compelling is that, following the 'Shared' setting scaling curve, the 'Crop+Resize' and 'Crop' settings' performance increases with longer pretraining, reducing the gap with the 'Original' one and converging to the same performance.

**Data augmentation and model capacity.** In Fig. 3 (right), we report ImageNet1k linear results when varying both data-augmentation strategies and model size. We can see that the gap between the best setup without hand-crafted data-augmentations and the classical setup ('Crop+Resize' or 'Crop' versus 'Original') increases when we use bigger model size, going from $-0.6\%$ for a ViT-S and ViT-B to $-1.1\%$ for a ViT-L and $-1.5\%$ for a ViT-H. This means that as we scale our model, we need more augmentation to increase the amount of data and reduce overfitting. This is in line with the most common understanding of scaling laws in experimental deep learning [29]: we need to scale compute, model size and data at the same time. This also explains why previous work [11, 39] had a hard time scaling self-supervised JEA without data-augmentations. They needed larger dataset, more compute and larger models, but all previous experimental studies on the impact of data-augmentations in SSL were mainly applied to ImageNet1k with smaller ResNets. This also explains why data-augmentation strategies became more aggressive during the evolution of self-supervised methods from its inception to modern methods like DINOv2, when people were scaling models but not datasets.

**Role of the 'reconstruction-based' iBOT loss.** While DINO and iBOT use the exact same loss, they differ in terms of target. With DINO, the teacher and the student are looking at two different views of the same image, with either different data augmentations in the 'Original' setting, the exact same in the 'Shared' one, or no augmentation at all with the 'Crop' one. In the iBOT case however, the student and the teacher see the same exact augmented view, but the student has some masked tokens.

Table 4: **Impact of the iBOT loss** on linear evaluation for multiple datasets for a ViT-L trained for 500 epochs on LVD-142M. We compare results with and without using masking and the iBOT loss.

|  |  | IN1k | | INat18 | | ADE20k | | NYU-Depth $\downarrow$ | |
|---|---|---|---|---|---|---|---|---|---|
|  | iBOT + masking | ✗ | ✓ | ✗ | ✓ | ✗ | ✓ | ✗ | ✓ |
| Data Aug. | Original | 84.0 | 84.4 | 74.7 | 73.6 | 41.4 | 45.6 | 0.430 | 0.413 |
|  | Shared | 82.6 | 84.0 | 74.1 | 73.6 | 40.7 | 44.8 | 0.448 | 0.414 |
|  | Crop+Resize | 82.6 | 83.5 | 71.3 | 71.1 | 40.3 | 44.4 | 0.437 | 0.415 |
|  | Crop | 82.1 | 83.4 | 69.8 | 71.4 | 40.6 | 43.5 | 0.445 | 0.417 |

It means that regardless of the data-augmentation strategy we use, both the teacher and the student see the same thing in the larger crops. Both settings are similar, and despite iBOT not using a conditional latent variable, it is possible to see it as a masked image modeling task in latent space, which is more akin to joint embedding predictive architecture (JEPA) than joint embedding architecture (JEA). To remove all confounding factors, we ablate the iBOT loss by setting its weight to zero and by removing the masking strategy in Table 4. We can see that it is still possible to train a very good model without iBOT and masking even if we use a stricter definition of JEA. The gap between the 'Original' and 'Crop' settings increases slightly, eg. from $1.0\%$ with iBOT to $1.9\%$ without, for ImageNet1k linear probing, or from $0.04$ to $0.015$ RMSE for NYUd, but the performances remain competitive; more importantly, it is in fact possible to train DINO alone without hand-crafted data-augmentations.

**Comparison against other models.** We compare our model trained on ImageNet22k with the 'Crop' setting against other SOTA models trained without hand-crafted augmentations in Table 1. It's worth noting that our setup uses the smaller amount of augmentations: AIM [34] uses 'Crop+Resize' and `RandomHorizontalFlip`, BEIT [4, 50] uses 'Crop+Resize' and `ColorJitter`, MAE [25] uses 'Crop+Resize' and I-JEPA [2] uses 'Crop+Resize'. Our model outperforms alternatives on a wide variety of tasks by a significant margin without hand-crafted data-augmentations.

### 4.3 Is scaling all you need in research?

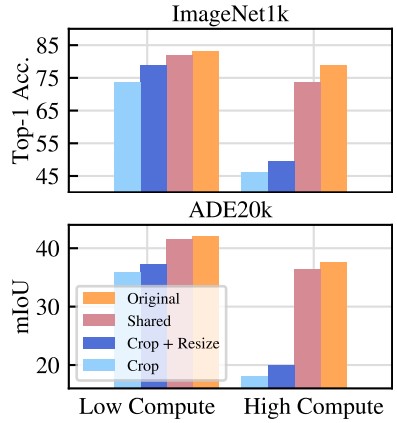

| Param. | IN1k | INat18 | ADE20k | NYUd $\downarrow$ |
|---|---|---|---|---|
| High Compute | 49.6 | 28.5 | 20.0 | 0.650 |
| Layer Decay | 47.6 | 26.8 | 20.7 | 0.693 |
| # Prototypes | 48.8 | 27.6 | 20.7 | 0.661 |
| Head Dim. | 49.0 | 27.8 | 17.7 | 0.654 |
| Momentum | 49.7 | 28.0 | 18.2 | 0.654 |
| Warmup Epoch | 51.3 | 26.0 | 24.4 | 0.605 |
| Warmup Temp | 51.3 | 28.5 | 22.1 | 0.627 |
| Drop Path Rate | 52.3 | 30.6 | 22.9 | 0.631 |
| Weight Decay | 53.4 | 30.5 | 21.7 | 0.646 |
| LR | 74.2 | 47.6 | 33.5 | 0.517 |
| Low Compute | 78.7 | 54.2 | 37.2 | 0.486 |

Figure 4: **(left): Impact of hyper-parameter optimisation's target compute** on the accuracy of a linear probe on ImageNet1k and ADE20k *for models trained on ImageNet1k*. We can see that optimising for high compute leads to poor performances on the 'Crop' and 'Crop+Resize' settings, which is the opposite of our findings when we optimize for low compute. **(right): Impact of hyperparameter tuning** using 'Crop+Resize' on ImageNet1k for 100 epochs. For each line, we switch only one hyper-parameter from the configuration optimized on ImageNet22k for 500 epochs (High compute) to the one optimized for ImageNet1k for 100 epochs (Low).

**Same experiment, different outcomes.** During this study, we used *two sets of hyper-parameters optimized for different use cases*. One optimized for training our models on the smaller scale dataset ImageNet1k during 100 epochs (low compute) that we used for all trainings on ImageNet1k, and one for training those models on the larger datasets ImageNet22k and LVD-142M during 500 epochs (high compute) that we used for all trainings on ImageNet22k and LVD-142M. Indeed, training on ImageNet1k using hyper-parameters optimized on the larger datasets gives different conclusions.

We show in Fig. 4 **(left)** the ImageNet1k linear-probe accuracy and ADE20k linear segmentation mIoU obtained after training a ViT-L on ImageNet1k when we use the 'Low Compute' and 'High Compute' sets of hyper-parameters. Interestingly, DINOv2 doesn't work when trained on ImageNet1k using the 'Crop' or 'Crop+Resize' settings with the 'High Compute' hyper-parameters. It reaches only $46\%$ linear accuracy, despite obtaining SoTA performances when trained on ImageNet22k and LVD-142M with the exact same hyper-parameters, providing nice scaling curves in the process. However, when we optimize the hyper-parameters for the specific scale of data (ImageNet1k) and compute (100 epochs), we can see that it reaches $73.5\%$ with a linear-probe, as presented before.

With the 'High Compute' hyperparameters, going from the 'Original' to 'Crop+Resize' setups makes DINOv2 lose 29.1% on the linear-probe evaluation, which is in phase with previous literature [39]. This means that if our only goal was to get a scaling curve leading to SoTA models, without optimizing for lower compute and data, we wouldn't have been able to disprove the assumption from previous studies that pretraining a strong JEA without data-augmentations with minimal loss in performances is impossible when using ImageNet1k [11, 39]).

**The issue with high-scale experimental studies.** In Fig. 4 (**right**), we show that it's not trivial to go from one hyper-parameter setup to the other, gains being non linear when tweaking parameters one at a time. Such issue already arose previously in the literature. For example, it was originally thought that ViT needed hundreds of millions of images to reach good performances [17], but such claim has been proven false in later work [20, 48]. The same happened with Large Language Models, with work scaling them to hundred of billions of parameters like OPT [54] reaching 175B before newer methods showed improved performances with smaller sizes as in [23, 47].

## 5 Conclusion

In this work, we challenged the common belief that joint-embedding architectures require data augmentations to deliver strong performance. The underlying interpretation of that belief, followed by theorists, is that the core learning signal in these methods relies on mapping differently augmented samples to the same embedding in the latent space. In particular, this reasoning implies expert knowledge in the design of data augmentations. Also, the photometric augmentations of images (such as blurring and color jittering) cannot be easily mapped to other modalities (such as speech or text). As such, this belief restricts the scope of high-performance JEA methods to images.

Our experiments show that this is not true, as it is possible to achieve performance similar to data-augmented pipelines with the DINOv2 SSL algorithm without using what is referred to, in prior literature, as *hand-crafted data augmentations*. Therefore, our results prove that joint-embedding architectures do not necessarily require domain knowledge during training to deliver state-of-the-art performance. The data augmentations merely influence training by increasing the dataset size, as shown by our 'Shared' setup, which does not enforce the learning of invariances.

Additional experiments with even weaker data augmentations, such as the 'Crop' setup, show that given enough training iterations and data, the models will converge to comparable performance as their stronger data-augmented counterparts. Our results suggest that prior conclusions largely depended on the smaller scale of experiments. However, we note that strong data augmentations lead to stronger performance in the case of shorter training, suggesting that they also help ease the optimization process during learning. We hope future work can shed more light on that observation.

**Limitations.** Given the cost of pretraining models, we focused on the leading self-supervised joint-embedding architecture, DINOv2, that provides strong performance at different data scales. While the results might be different for other algorithms, in principle, the existence proof that we offer in this work still holds. We noted that data augmentations tend to ease optimization, but the underlying mechanism is not known at this point. We merely claim that handcrafted photometric augmentations are optional for successful learning. While it has a measurable effect on some setups, this does not invalidate our conclusions. For large architectures, there is still a tiny gap in performance between augmented and non-augmented setups for long training. While we hypothesize that this might be due to slight variations in the optimal hyperparameters, there could also be a deeper reason that is not covered within the scope of this study.

**Statement of Broader Impact.** The total compute cost for this study was approximately 150k GPU-hours. We used strong models to blur all human faces in our web-based image data pool.

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
