# OpenReview forum: "You Don’t Need Domain-Specific Data Augmentations When Scaling Self-Supervised Learning"
_NeurIPS.cc/2024/Conference — NeurIPS 2024 poster_

### Official Review · Reviewer_PAfm · 2024-06-21

**Soundness:** 3
**Presentation:** 4
**Contribution:** 3
**Rating:** 7
**Confidence:** 4

**Summary:**

This paper examines the question of whether data augmentations, specifically hand-crafted domain-specific augmentations, are needed for self-supervised learning.  This question has relevance as the existing state of the art methods use such augmentations in the standard natural image domain, which leaves open the question of what one should do when trying to apply self-supervised learning to a new domain, eg medical imaging.

Through experimental analysis, this paper demonstrates that the primary impact of data augmentation is in increasing the effective dataset size, and that with sufficient data, compute, and model complexity, the gap between performance with and without such data augmentations can be largely closed.  In particular, the paper shows state of the art results under the condition of not using hand-crafted augmentations, while using a joint-embedding architecture.

**Strengths:**

+ paper is well-written and well-reasoned, making it easy to follow
+ the question examined is of interesting fundamental importance, in giving insight into self-supervised learning and what one should take into account when applying SSL to new data modalities
+ convincing experimental results to support the argument that the main effect of data augmentation in SSL is for increasing effective training set size, and that by scaling data set size, the performance gap with and without hand-crafted augmentations closes

**Weaknesses:**

- the experiments were all using DINOv2 - this is a secondary weakness since the paper offers the results as an existence proof that hand-crafted augmentations are not necessary.  however, to have a more complete understanding of the effect of such hand-crafted augmentations, it would be interesting to know whether the results found hold in general for SSL, even if the specific learning algorithm/architecture, were changed, to know whether this is a general phenomenon or if there is an aspect that is specific to something in DINO.

**Questions:**

As a more open-ended question, do your results have any implications for reconstruction-based methods for SSL?  Ie, in looking at Table 1, considering the proposed method is able to achieve better results than existing reconstruction-based methods, while being similarly constrained to not leverage hand-crafted augmentations, do you have thoughts on why this gap existing, or put another one, what kind of information would need to be added to the reconstruction-based methods to close the gap?

**Limitations:**

yes

---

> ### Author Rebuttal · Authors · 2024-08-07
>
> We thank the reviewer for their positive reception of our work. In the following we will be addressing each of your comments.
>
> **W1**) Please refer to the general rebuttal. We add an analysis on MOCOv3 (a more modern version of the SimCLR paper) and explain why our choice is constrained to DINOv2 (the Crop setup was unstable with MOCOv3 and we didn't manage to train a model with it). Our claim might hold for other methods as long as it’s possible to scale them effectively to ImageNet-22k (which is not an easy task). In the meantime, we think that our paper can already impact people training DINOv2 models on other modalities [1,2,3,4,5,6].
>
> **Q1**) It’s an interesting question about insights on reconstruction-based SSL. Our paper shows that the constraint on hand-crafted data-augmentations is not the main factor in the existing gap, but multiple factors can still explain it. First, our model uses DINOv2, which is carefully optimized to work on large scale datasets. I-JEPA, however fails to train good models on ImageNet-22k, but according to the results in our paper, this could be explained by the use of suboptimal hyper-parameters. MAE also fails to scale effectively, in line with a new paper explaining that learning by reconstruction forces the model to learn uninformative features [7]. At the end, the issue might be that we add too much unused information to learn for reconstruction-based methods, and that scaling data increases even more this amount of information.
>
> [1] R. J. Chen et al. Towards a general-purpose foundation model for computational pathology
> [2] H. Xu et al. A whole-slide foundation model for digital pathology from real-world data
> [3] E. Vorontsov et al. Virchow: A Million-Slide Digital Pathology Foundation Model
> [4] T. Moutakanni et al. Advancing human-centric AI for robust X-ray analysis through holistic self-supervised learning
> [5] F. Perez-Garcia et al. RAD-DINO: Exploring Scalable Medical Image Encoders Beyond Text Supervision
> [6] J. Tolan et al. Very high resolution canopy height maps from RGB imagery using self-supervised vision transformer and convolutional decoder trained on aerial lidar
> [7] R. Balestriero et Y. LeCun, Learning by Reconstruction Produces Uninformative Features For Perception

---

> > ### Comment · Reviewer_PAfm · 2024-08-13
> >
> > Thank you for your detailed response.

---

> > > ### Author Response · Authors · 2024-08-14
> > >
> > > We thank the reviewer for the positive feedback. We will include the discussion to the revised manuscript.

---

### Official Review · Reviewer_Jk8X · 2024-07-06

**Soundness:** 3
**Presentation:** 3
**Contribution:** 3
**Rating:** 5
**Confidence:** 4

**Summary:**

Traditionally, it is believed that the effectiveness of Joint-Embedding Architectures (JEAs) such as SimCLR lies in their ability to map augmented views of the same image to the same representation in the latent space, thus requiring specific data augmentations that lead to superior downstream performance.  Moreover, such augmentations, like blurring and color jittering, are not directly applicable to non-visual data like speech or text, potentially limiting the utility of JEAs to image-only applications.

This work challenges the necessity of hand-crafted data augmentations for training Self-Supervised Learning (SSL) models with JEAs. Specifically, the authors provide a rigorous study using DINOV2, demonstrating that effective image representations can be achieved using minimal augmentation—specifically, only cropping without resizing—as long as the dataset is sufficiently large. Further, augmentations mainly serve to expand the effective size of the training dataset rather than enforcing the learning of invariances. Role of augmentations is studied on the three critical scaling dimensions in deep learning: data volume, computational power, and model size. To support these claims, experiments are conducted across a range of dataset sizes, from smaller datasets like ImageNet1k to larger ones such as LVD142, and included multiple model sizes, from small to large.

**Strengths:**

**Originality.** A lot of previous works focus on studying the impact of different data augmentations both from a theoretical and empirical perspective. This paper additionally studies this impact at scale, which to the best of my knowledge, is completely novel. It is clear how this work differs from previous contributions, and the analysis shows new insights.

**Quality.** The paper is well written and the claims are mostly well-substantiated with extensive experimentation supporting them.

**Clarity.** The paper is well-organized and clear, with understandable figures. Different data augmentation approaches and hyperparameter choices are clearly outlined.

**Significance.**
- The work provides evidence that one could train state of the art SSL approaches with *DINOv2* without on hand crafted data augmentations except cropping and resizing. This could be useful for scaling SSL applications to non-image domains such as time series etc.
- The idea of studying the use of data augmentations to enforce invariance through “Shared” augmentation setting is very interesting
- The role of compute and non linear gains with changing hyperparameters is important for large scale training and is well highlighted

**Weaknesses:**

1.  My major concern is with respect to the usefulness and impact of this work. The authors highlight training of large scale SSL models with just cropping and resizing. As also mentioned in the paper, the key use case of this study would be generalizability of SSL approaches to domains such as medical imaging with totally different channels and characteristics. However, cropping may not be an effective augmentation strategy in these domains—for instance, in medical imaging, key features like cells or tumors often occupy a small portion of the image. The paper lacks evidence with respect to their stated motivation– Does their learnt representation actually generalize better to non-vision domains as compared to the representations learnt through traditional hand crafted augmentations ?

2. Despite using data augmentations such as color jitter in JEPAs, it is unclear if the model is actually learning invariance to them. In some works [1][2], it has been shown that despite training a predictor in the latent space, the model can ignore the augmentations and merely learn invariance. There is no discussion from this perspective, which I believe is very relevant to the work.

- [1] Garrido, Quentin, Laurent Najman, and Yann Lecun. "Self-supervised learning of split invariant equivariant representations." arXiv preprint arXiv:2302.10283 (2023).

- [2] Gupta, Sharut, et al. "In-Context Symmetries: Self-Supervised Learning through Contextual World Models." arXiv preprint arXiv:2405.18193 (2024).

3. I would like to see more insights into the gaps in performance observed in low and large data regimes. By the claims made in the paper, with long enough training, training without data augmentations should achieve the same performance as its counterpart. However, a gap still remains. More clarity on this would benefit the work.

I am willing to change my score if the authors address the above comments.

**Questions:**

1. What is “AIM” in Table 1? The abbreviations for algorithms used in Table 1 are not defined or mentioned in the text.

2. “DFN-2B+” is undefined and used in Table 1. What is it?

**Limitations:**

Mentioned in the Weaknesses section

---

> ### Author Rebuttal · Authors · 2024-08-07
>
> We would like to thank you for recognizing the importance of scaling SSL on other domains. In the following we extended our experiments, including also other imaging domains, making our conclusions more solid.
>
> **W1**)  You indicated that the main motivation for this work, which is to allow using SSL models on different domains, was not supported by experimentation. Indeed, generalization to new domains is very important given that multiple papers used DINOv2 on medical imaging, remote sensing, and many other ones, either directly as is or as a pretrained model to finetune [1,2,3,4].  We decided to test (Table 1 in the rebuttal) our models on such domains where invariance toward domain-specific augmentations could be harmful:
> - logos, signs and product’s packaging where for example text can be degraded by blurring and zooming, and arrows can be inverted if the features are invariant to the flipping;
> - remote sensing, where colors and zooming can have a big impact on the performances;
> - medical imaging where classes differ in type (shape vs texture) and size (small nodule, broken ribs, enlarged heart etc.).
>
> Table 1 in the rebuttal shows that a model trained without domain-specific augmentations tends to generalize better on most of these domains, notably on products (RP2K and Products-10k), remote sensing (FireRisk) and medical dataset (MIMIC, CheXpert, Vindr and NIH-14).
>
> We gained 4.8% acc. on GTSRB when switching from the original DINOv2 to the Crop only one, showing the impact of resizing on this task.
>
> On RP2K, Products-10k, and medical imaging datasets, all methods without domain-specific data-augmentations get, in general, better performances, proving that those augmentations were tailored for ImageNet-like tasks but can be detrimental to other ones, as stated in [8].
>
> **W2**) Thank you for the comment. Indeed, models might not learn invariance even when trained with data-augmentations. However, we quantitatively show in Table 3 in the rebuttal that the Original setup is more invariant than the Shared, Crop+Resize and Crop setup toward data-augmentations. For that, we follow the prior work of [5, 6] and use as a metric the average cosine similarity between augmented views of the same image. We also normalize based on the average similarity between random images to take into account potential model variability.
>
> Analyzing the quantitative results of our empirical study in Table 3, we observe an interesting pattern, with the invariance decreasing between the Original,  Shared, Crop+Resize and Crop settings. It was expected that the most invariant setup would be “Original”, and the least invariant would be “Crop”, but it’s interesting to see that the Shared and Crop+Resize setup have different levels of invariance despite having the same “invariance enforcement” in the loss. A more detailed analysis of data augmentations and equivariance would be a good follow-up to this paper.
>
> **W3**) The gap between low and large data regimes cannot be recovered only by training for longer on the smaller dataset. As it is discussed in this study, scaling SSL is harder than only changing the dataset’s size. Increasing the number of epochs alone will lead to overfitting. This happens less with data-augmentations as they artificially increase the dataset size. Increasing the number of data alone will help, but the training will be suboptimal as the model will see each image for a shorter amount of time.
> Thus, scaling both training data and length is what can make models with and without data augmentations reach the same performances.
>
> Another remark about the remaining gap for our 500 epochs setup with LVD-142M with and without augmentations. We shortly mentioned this in the limitations, but here is a more detailed explanation:
> - Hyperparameters were tuned by DINOv2’s author using all data-augmentations. We used the exact same parameters for all our setups, and thus are probably using suboptimal parameters when removing augmentations.
> - The larger the model used, the more data we need to close the gap. You can see this phenomenon in the paper, Figure 3 on the right. We might be in a setup (ViT-L)  where more data is needed to fill the gap (for example, CLIP ViT-L models are trained using billions of images [7]).
> - We don’t use data-augmentations in our linear classifier training to remove confounding factors when evaluating our models. Adding data-augmentations here isn’t necessary to reach best performances using the original DINOv2. However, some invariances are actually good on ImageNet-like domains [8], and the ones used in the original setup are tailored exactly for this. The gap might close even more if we optimized data-augmentations during the classifier training, transferring the invariances from the pretraining step to the classification step.
>
> Questions:
> Sorry for the confusion about AIM and DFN-2B+. These abbreviations come from the paper [9]. We will add all citations to Table 1.
>
>
> [1] X. Song, General Purpose Image Encoder DINOv2 for Medical Image Registration
>
> [2] M. Baharoon et al. Towards General Purpose Vision Foundation Models for Medical Image Analysis: An Experimental Study of DINOv2 on Radiology Benchmarks
>
> [3] B. Cui et al. Surgical-DINO: adapter learning of foundation models for depth estimation in endoscopic surgery
>
> [4] X. Bou et al. Exploring Robust Features for Few-Shot Object Detection in Satellite Imagery
>
> [5] Q. Garrido et al. Learning and Leveraging World Models in Visual Representation Learning
>
> [6] A. Devillers & M. Lefort, EQUIMOD: an equivariance module to improve visual instance discrimination
>
> [7] M. Cherti, Reproducible scaling laws for contrastive language-image learning
>
> [8] I. Bendidi et al. No Free Lunch in Self Supervised Representation Learning
>
> [9] A. El-Nouby et al. Scalable Pre-training of Large Autoregressive Image Models

---

> > ### Comment · Reviewer_Jk8X · 2024-08-10
> >
> > Many thanks for conducting additional experiments and providing a detailed rebuttal that addresses many of the weaknesses identified and questions raised. I appreciate the addition of results on medical imaging, remote sensing and logos and signs datasets. I emphasize that all clarifications made during this rebuttal should be added to the revised manuscript to improve clarity of the work.
> >
> > Given the rebuttal addressed most of my concerns, I have increased my score to 5.

---

> > > ### Author Response · Authors · 2024-08-14
> > >
> > > We thank the reviewer for the positive feedback and valuable suggestions that helped us improve our paper. We will of course include the contents of the rebuttal in the camera ready.

---

### Official Review · Reviewer_hWoD · 2024-07-19

**Soundness:** 2
**Presentation:** 4
**Contribution:** 3
**Rating:** 5
**Confidence:** 5

**Summary:**

This paper demonstrates the possibility of training JEA SSL encoders using limited augmentations (not 0 augmentations as in the title) compensated by significantly increasing the size of the training data. This is built on the claim that most SSL works have relied on augmentations for SOTA performance. The experiments show that training with crop+resize augmentation alone, with much larger training data can match the performance of ImageNet-1K + hand-crafted augmentations. More experiments demonstrate the benefit of scale and how augmentations' role diminishes as scale increases.

**Strengths:**

- The paper is very well written and easy to follow. Motivation and experimental results/takeaways are presented in a clear format.
- The case study, while only focusing on DINOv2 and iBOT, is thorough with enough supporting ablations to the claims
- Discussion on the role of scale throughout the paper is appreciated and certainly useful to conduct further research

**Weaknesses:**

- Misleading title - The title "you don't need data augmentations" is certainly misleading since cropping (an augmentation) is still applied under a large dataset regimes. The authors have also not demonstrated results on other methods to make a general claim about the un-usefulness of augmentations in SSL as a whole.

- (Crop + Resize) and (Crop) are both still augmentations - I am skeptical about the main claims because I disagree with the main assumption that "crop" and "crop+resize" (1) do not increase the dataset size and (2) do not induce invariance. In "Crop" randomly removing 32 pixels (256 - 224) from the height and width, very much changes the spatial information compared to the original image and certainly induces an invariance when used in multi-view SSL. Have the authors studied the impact of reducing cropping strength? What happens if you resize to say 230 and crop to 224?
With the above concern in mind, Crop+Resize is an even stronger augmentation and surely increases the dataset size artificially which questions several explanations of results (for example - line 220). A simple visual examination of RandomResizedCropped ImageNet-1K samples could show several low-overlapping copies of the same image since the main content need not always be a single object at the center of the image.

- Masking is also an augmentation - Similar to the above discussion on cropping, masking is also a form of augmentation because it fundamentally alters input information and training in such a manner forces an invariance to such alterations.

- Need results on other models - While the studies on DINOv2 are extensive, the bold claim of not needing augmentations for SSL needs to be studied across the realm of SSL because it contradicts the SimCLR paper (as discussed by authors in Introduction section) and a recent work [1].

[1] Morningstar et al, Augmentations vs Algorithms: What Works in Self-Supervised Learning

**Questions:**

See weaknesses section

**Limitations:**

Discussed but more raised in the Weaknesses section.

---

> ### Author Rebuttal · Authors · 2024-08-07
>
> We would like to thank the reviewer for recognizing the importance of the role of scale in self supervised learning and for their questions that made us think more deeply about the semantics and experiments we used throughout our paper. In the following we will be addressing each of your comments.
>
> **W1**) Please refer to the general rebuttal. Thanks to your comment, the title of our paper will be changed to: “Self-supervised learning doesn’t need domain-specific data augmentations at scale” to highlight more the claims of the paper.
>
> **W2-3**) Hand-crafted data-augmentation is the term used in the original I-JEPA paper [1] to talk about most augmentations but masking and RandomResizedCrop. Hand-crafted data augmentations are mentioned in I-JEPA as augmentations that can’t be used on other modalities (thus hand-crafted for the specific modality in use). In our paper, we used the same term but included the Resizing as a hand-crafted augmentation as it isn’t usable in text or modalities where resampling is impossible or doesn’t make sense.
>
> We propose to change “hand-crafted” to “domain-specific” to make it more explicit. We think that this change in semantics (we used hand-crafted data augmentation in the paper but not the original title) answers both of your points about the boldness of our claim (that was, we think, only in the title) and your question about the fact that cropping and masking are augmentations. We will also make sure to update the paper with the correct wording when necessary.
>
> To support our claims, we actually don't need our model to be trained with *exactly* zero invariance or data augmentations. A lot of papers are claiming that data augmentations are necessary to achieve good performance with classical SSL [1,2,3] and that it’s not the case for reconstruction-based methods like MAE, I-JEPA or AIM [4,5]. However, MAE, I-JEPA, and AIM still use data augmentations (masking and RandomResizedCrop). It’s also worth noting that all the data-augmentation ablations in [1] are only done on what we call “domain-specific augmentations” and that all their setup include RandomResizedCrop. We show that their results don't hold at scale with DINOv2, which we think is novel and interesting.
>
> We claim that we can remove all domain-specific augmentations and still train a very powerful model, using fewer augmentations than all the other methods in the literature.
> When we analyze the impact of data-augmentation and invariance (Table 3 in the rebuttal), we don’t need it to be exactly zero for our claims to hold. We just need the Shared and Crop+Resize setup to enforce less invariance than the Original setup. The performance gap that is created by enforcing less invariance is compensated by increasing the amount of data, showing that the main bottleneck was the artificial increase of data, not the invariance making. The impact of the RandomCrop and masking augmentations doesn’t change between all our setups, and we only analyze the difference relative to all the other augmentations.
>
> We also want to add that we have a table in the original paper (Table 2) that ablates the masking augmentation, training a strong model with only one augmentation (RandomCrop, without resizing and without masking). This model still beats all the other baselines from Table 1 in our paper by a large margin.
>
> **W4**) Please refer to the general rebuttal. We add an analysis on MOCOv3 (a more modern version of the SimCLR paper) and explain why our choice is constrained to DINOv2 (the Crop setup was unstable with MOCOv3 and we didn't manage to train a model with it). Our claim might hold for other methods as long as it’s possible to scale them effectively to ImageNet-22k (which is not an easy task). In the meantime, our paper can already impact people training DINOv2 models on other modalities [6,7,8,9,10,11].
>
>
> [1] Morningstar et al, Augmentations vs Algorithms: What Works in Self-Supervised Learning
>
> [2] T. Chen et al. A Simple Framework for Contrastive Learning of Visual Representations
>
> [3] J.-B. Grill et al. Bootstrap your own latent: A new approach to self-supervised Learning
>
> [4] M. Assran et al. Self-Supervised Learning from Images with a Joint-Embedding Predictive Architecture
>
> [5] A. El-Nouby et al. Scalable Pre-training of Large Autoregressive Image Models
>
> [6] R. J. Chen et al. Towards a general-purpose foundation model for computational pathology
>
> [7] H. Xu et al. A whole-slide foundation model for digital pathology from real-world data
>
> [8] E. Vorontsov et al. Virchow: A Million-Slide Digital Pathology Foundation Model
>
> [9] T. Moutakanni et al. Advancing human-centric AI for robust X-ray analysis through holistic self-supervised learning
>
> [10] F. Perez-Garcia et al. RAD-DINO: Exploring Scalable Medical Image Encoders Beyond Text Supervision
>
> [11] J. Tolan et al. Very high resolution canopy height maps from RGB imagery using self-supervised vision transformer and convolutional decoder trained on aerial lidar

---

### Author Rebuttal · Authors · 2024-08-07

We thank all the reviewers for their insightful questions.

To recapitulate our paper, we summarize our contributions:
- We can train a powerful and scaled self-supervised model with no domain-specific data-augmentations, in contrast to all alternative approaches. This model provides an existence proof that it is possible to train a model with a similar performance to its domain-specific data-augmented counterpart. This is true only at the ImageNet-22k scale.
- We reproduce the results of prior literature [1,2,3] at the ImageNet-1k scale, showing that domain-specific data augmentations are necessary (if the amount of data is too small).
We conclude that SSL does not rely on enforcing representational invariance to domain-specific data augmentations in order to perform well: it is possible to obtain strong models (within 1% of the state-of-the-art) as long as we scale training data.

We never claimed that data-augmentations were un-useful, nor that removing them would lead to better performances on all benchmarks (we actually get a performance boost on some domains and show this in rebuttal table 1). In this work, we discuss that we don’t need most of them to get great performances at scale, disproving the belief that they are the key to modern SSL.

Given that our title might be less specific than the claims in our paper, we will rename our article “Self-supervised learning doesn’t need domain-specific data augmentations at scale”.

Our conclusion doesn’t contradict but extends the prior knowledge present in the SSL literature (e.g. the SimCLR work [2], and more recent work such as Morningstar et al. [1]), proving that enforcing domain-specific invariances is not the core learning mechanism of SSL, and that dataset scale was the issue. The data augmentations that we use, i.e. cropping and masking, are very general by nature, suggesting it is possible to extend SSL approaches (in particular DINOv2) to other domains, given proper scale, while data-augmentation can also be harmful to specific domains or classes [4].

The reviewers notably complained that our experiments were limited to only the DINOv2 and DINO methods. There are several reasons for this:
- Our claim about not needing domain-specific augmentations only works with larger scale datasets, but to the best of our knowledge, DINOv2 is the only method that scales to the ImageNet-22k scale. We provide additional experiments in Table 2 of the rebuttal, where we trained MOCOv3 [5] (an improved version of SimCLR) on ImageNet-1k and -22k, with and without domain-specific data augmentations. First, we see that we obtain the same results as DINOv2 when training on ImageNet-1k: removing augmentations has a big impact. We also observe that the method does not scale well to more data - at least not with the hyperparameters provided by the authors - and that ImageNet classification performance decreases when using the larger dataset. The same scaling issue happens with the official I-JEPA models [6] that we tested in Table 1 of our paper.
- Scaling SSL to more data is harder than just changing the dataset. Tweaking hyperparameters to scale up a method is extremely compute-intensive (3-6M GPU-hours for DINOv2 [7]), and we believe that this is not necessary to support the claim that SSL does not need domain specific augmentations to get good performances: a single existence proof, that we provide in this work, appears sufficient. Given the popularity of DINOv2 as the state-of-the-art self-supervised methods on different domains [8,9,10,11,12,13], we think our work can have an important impact on more powerful and scaled SSL methods. In the Rebuttal Table 1, we analyze the generalization capabilities of our different data-augmentation setups and show that removing augmentations can be beneficial in some domains.

We also add results in Rebuttal Table 3 to quantify the invariance toward augmentation between the different models, proving that removing data-augmentations indeed reduce the learned invariance.


[1] Morningstar et al, Augmentations vs Algorithms: What Works in Self-Supervised Learning

[2] T. Chen et al. A Simple Framework for Contrastive Learning of Visual Representations

[3] J.-B. Grill et al. Bootstrap your own latent: A new approach to self-supervised Learning

[4] I. Bendidi et al. No Free Lunch in Self Supervised Representation Learning

[5] X. Chen et al. An Empirical Study of Training Self-Supervised Vision Transformers

[6] M. Assran et al. Self-Supervised Learning from Images with a Joint-Embedding Predictive Architecture

[7] M. Oquab et al. DINOv2: Learning Robust Visual Features without Supervision

[8] R. J. Chen et al. Towards a general-purpose foundation model for computational pathology

[9] H. Xu et al. A whole-slide foundation model for digital pathology from real-world data

[10] E. Vorontsov et al. Virchow: A Million-Slide Digital Pathology Foundation Model

[11] T. Moutakanni et al. Advancing human-centric AI for robust X-ray analysis through holistic self-supervised learning

[12] F. Perez-Garcia et al. RAD-DINO: Exploring Scalable Medical Image Encoders Beyond Text Supervision

[13] J. Tolan et al. Very high resolution canopy height maps from RGB imagery using self-supervised vision transformer and convolutional decoder trained on aerial lidar

---

### Decision · Program_Chairs · 2024-09-25

**Decision:**

Accept (poster)

**Comment:**

The paper, originally titled "You Don’t Need Data Augmentations in Self-Supervised Learning," addresses a important question in the field of Self-Supervised Learning (SSL): whether strong, hand-crafted data augmentations are essential for effective model training, particularly in Joint-Embedding Architectures (JEAs). Running empirical simulations with the DINOv2 model, the authors show that effective image representations can be achieved with minimal augmentation (eg. cropping without resizing) provided the training dataset is sufficiently large. The study uses a range of dataset sizes, from ImageNet1k to the large-scale LVD142 dataset, and studies models of various sizes. The results indicate that augmentations primarily serve to expand the effective dataset size rather than enforce invariance learning, especially when scaling up.

The authors clarified during the rebuttal period that they do not claim data augmentations are universally unnecessary. Instead, they argue that at scale, most augmentations are not required to achieve strong performance. The minimal augmentations used in this study, i.e. cropping and masking, are general by nature, suggesting that SSL approaches could be extended to other domains without relying on domain-specific augmentations. This point is particularly relevant for fields like medical imaging, where such augmentations may not be well-defined or applicable. The authors propose a more accurate title for the final version: "Self-supervised learning doesn’t need domain-specific data augmentations at scale."

The paper acknowledges that removing data augmentations may not lead to improved performance across all benchmarks. However, it provides valuable insights into the role of augmentations in large-scale SSL models. The findings suggest that with sufficient data and model complexity, the gap in performance between models trained with and without domain-specific augmentations can be significantly reduced.

In summary, this paper offers an interesting and useful  contribution to the current understanding of SSL methods by challenging the conventional view regarding the necessity of data augmentations. The reviewews recommend accepting the paper. The proposed title revision is also supported, as it more accurately reflects the scope of the work.